# A Novel *Rhodotorula evergladensis* CXCN-6 Rich in Torularhodin and PUFAs with Potent Antioxidant and Anti-Inflammatory Activities

**DOI:** 10.3390/antiox14121420

**Published:** 2025-11-27

**Authors:** Chaiwoo Park, Myeongsam Park, Tingting Li, Chunxiao Shen, Zuxuan Zheng, Yitong Ge, Xuanyan Jin, Maolin Wei, Jaehwan Choi, Jae Sung Hwang, Zhengqun Li

**Affiliations:** 1R&I Center, COSMAX China, 529 Xiaonan Road, Shanghai 201400, China; cwpark@cosmax.com (C.P.); msampark@cosmax.com (M.P.); tingting.li@cosmax.com (T.L.); chunxiao.shen@cosmax.com (C.S.); zuxuan.zheng@cosmax.com (Z.Z.); yitong.ge@cosmax.com (Y.G.); hy_kim@cosmax.com (X.J.); maolin.wei@cosmax.com (M.W.); jhchoipro@cosmax.com (J.C.); 2Department of Genetics & Biotechnology, Graduate School of Biotechnology, College of Life Sciences, Kyung Hee University, Youngin 17104, Republic of Korea

**Keywords:** *Rhodotorula evergladensis*, carotenoids, torularhodin, microbial lipids, antioxidant, anti-inflammatory

## Abstract

Carotenoids and microbial lipids are valuable bioactive compounds with potent antioxidant and anti-inflammatory properties. Microbial biosynthesis provides a sustainable alternative to conventional plant extraction and chemical synthesis. Here, we report the isolation and characterization of a novel oleaginous yeast, *Rhodotorula evergladensis* CXCN-6, from the surface of Nymphaea ‘Gorgeous Purple’. The strain displayed intense reddish-orange pigmentation due to intracellular carotenoid accumulation. HPLC and LC–MS analyses identified torularhodin as the predominant carotenoid (*m*/*z* 563.4 [M]^+^), while lipids were rich in linoleic (C18:2), oleic (C18:1), and α-linolenic (C18:3) acids. Under optimized fermentation, CXCN-6 yielded 63.56 mg/L torularhodin and 9.83 g/L total lipids. The CXCN-6 extract showed strong DPPH and ABTS radical scavenging activities and significantly reduced intracellular ROS levels in UVA (9 J/cm^2^)-stimulated HaCaT cells. It also suppressed IL-6 and TNF-α secretion in LPS-activated macrophages without cytotoxicity. Collectively, these results establish *R. evergladensis* CXCN-6 as a novel and efficient microbial platform for the co-production of torularhodin and PUFA-rich lipids with potent antioxidant and anti-inflammatory properties, offering promising applications in nutraceutical, skincare, and functional food industries.

## 1. Introduction

Carotenoids and lipids are two important classes of bioactive metabolites that play essential roles in both cellular physiology and human health. Carotenoids are naturally occurring pigments synthesized by plants, algae, fungi, and numerous microorganisms [1]. These compounds are responsible for the yellow, orange, and red hues in biological systems and are well known for their potent antioxidant and health-promoting properties [2]. Among microbial carotenoids, torularhodin—a linear, carboxylated carotenoid with an extended conjugated polyene chain—has recently drawn significant attention for its exceptional antioxidant, anti-inflammatory, and photoprotective activities [3,4,5]. Compared to β-carotene, torularhodin displays stronger singlet oxygen quenching capacity, greater oxidative stability, and enhanced biological activity, which together underscore its potential as a multifunctional natural compound for food, cosmetic, and pharmaceutical applications [6,7]. Although torularhodin shows encouraging bioactive properties, its broader commercial development remains limited, in part because comprehensive studies on cytotoxicity, toxicology, and clinical safety are still needed. As global interest in natural pigments and functional bioactive ingredients continues to rise, identifying new microbial producers with strong torularhodin biosynthetic potential will be important for supporting future industrial use and facilitating more thorough safety evaluations.

In parallel, microbial lipids—commonly referred to as single-cell oils (SCOs)—have emerged as sustainable alternatives to plant- or animal-derived oils [8,9]. These lipids, mainly composed of triacylglycerols enriched with saturated and unsaturated fatty acids, are valuable feedstocks for biofuels, functional foods, and health products [10,11]. Importantly, lipids enhance the solubility and stability of hydrophobic antioxidants such as torularhodin, offering synergistic potential in formulation science [12,13]. The co-production of torularhodin and lipids within a single microbial platform presents clear bioprocessing advantages—enhanced productivity, simplified downstream extraction, and improved cost-effectiveness—thus aligning well with the goals of the circular bioeconomy [14,15].

Traditional sources of carotenoids and lipids rely heavily on plant extraction or chemical synthesis, but these approaches are hampered by long growth cycles, seasonal and land-use constraints, and the need for hazardous reagents [16]. In contrast, microbial fermentation offers a more sustainable, scalable and controllable route for producing natural bioactives with high purity and reproducibility and allows targeted enhancement via metabolic or process engineering [17,18]. Among microbial sources, yeasts of the genus *Rhodotorula* are particularly noteworthy for their ability to synthesize both carotenoids—primarily torularhodin, torulene, and β-carotene—and intracellular lipids under stress or nutrient-limited conditions [19,20]. These traits make them promising microbial chassis for biotechnological applications across the food, healthcare, and cosmetic industries [21]. However, the diversity within this genus remains underexplored. While *Rhodotorula mucilaginosa*, *R. glutinis*, and *R. toruloides* have been extensively studied [17,22,23], *Rhodotorula evergladensis* is a lesser-known and previously uncharacterized species with unknown biosynthetic potential. To our knowledge, there have been no systematic studies investigating the metabolic, antioxidant, or anti-inflammatory properties of *R. evergladensis*. Exploring such non-model species may reveal unique biosynthetic pathways for high-value carotenoids like torularhodin and enrich the microbial platforms available for sustainable production.

In this study, we report the isolation and functional characterization of a novel oleaginous yeast strain, *Rhodotorula evergladensis* CXCN-6, from the surface of Nymphaea ‘Gorgeous Purple’. The strain exhibited intense reddish-orange pigmentation due to intracellular torularhodin accumulation, along with substantial lipid storage. Through combined spectrophotometric, chromatographic, and functional assays, we demonstrate that CXCN-6 co-produces torularhodin and unsaturated lipids with strong antioxidant and anti-inflammatory activities. These findings not only uncover the functional potential of *R. evergladensis* but also establish it as a promising microbial chassis for the sustainable production of natural multifunctional ingredients.

## 2. Materials and Methods

### 2.1. Sample Collection and Strain Isolation

*R*. *evergladensis* CXCN-6 strain was isolated from the Nymphaea ‘Gorgeous Purple’ of Yunnan-Kweichow Plateau, Kunming, China. The strain collection number of *R. evergladensis* CXCN-6 is recorded in the China General Microbiological Culture Collection Center (Beijing, China) as CGMCC 7.610. Nymphaea samples were collected in sterile tube. Samples were plated in yeast extract peptone dextrose (YPD) medium (Yeast extract 10 g/L, Peptone 20 g/L, Dextrose 20 g/L, Agar 15 g/L) and incubated at 28 °C for 2 days. The most reddish-orange colony was purified by five streaking rounds on fresh YPD medium and incubated at 28 °C for 48 h to obtain pure culture. Obtained strains were grown in a YPD medium and incubated at 28 °C for 48 h on rotary shaker at 150 rpm. The purified strains were preserved in 20% glycerol (*w*/*v*) at −80 °C for further use.

### 2.2. Strain Identification

The selected microorganism was identified based on the internal transcribed spacer (ITS) region of the ribosomal RNA gene. Genomic DNA was extracted using the TIANamp Yeast DNA Kit Gene (Tiangen Biotech (Beijing) CO., LTD., Beijing, China) according to the manufacturer’s protocol. The ITS region was amplified by polymerase chain reaction (PCR) using universal primers ITS1 (5′-TCCGTAGGTGAACCTGCGG-3′) and ITS4 (5′-TCCTCCGCTTATTGATATGC-3′). PCR reactions were performed in a total volume of 50 μL using a Bio-Rad T100 thermo-cycler (Bio-Rad Laboratories, Hercules, CA, USA), containing 2 μL of genomic DNA, 25 μL 2 × T8 High-Fidelity Master Mix, 2 μL of 10 μM each primer, 19 μL of sterile ultrapure water. The thermal cycling conditions were as follows: initial denaturation at 98 °C for 10 s, followed by 35 cycles of denaturation at 98 °C for 10 s, annealing at 55 °C for 10 s, and extension at 72 °C for 30 s, with a final extension at 72 °C for 5 min. The PCR products were verified by electrophoresis on a 1% agarose gel and subsequently purified and sequenced by Tsingke Biotechnology Co., Ltd. (Shanghai, China). The obtained ITS rRNA gene sequence (542 bp) was analyzed using the BLASTn on the NCBI online server (https://blast.ncbi.nlm.nih.gov) algorithm against nucleotide sequences available in the NCBI GenBank and EMBL databases to determine the closest phylogenetic relatives.

### 2.3. Phylogenetic Analysis

The ITS sequence of *Rhodotorula* sp. CXCN-6 was aligned with reference sequences of closely related *Rhodotorula* species retrieved from the NCBI GenBank database. Multiple sequence alignment was conducted using ClustalW implemented in MEGA 7 software (version 7.0.26) [24]. A phylogenetic tree was constructed using the neighbor-joining (NJ) method, with 1000 bootstrap replications to assess the statistical support of each clade. Evolutionary distances were computed using the Kimura 2-parameter model. The resulting phylogenetic tree was visualized and annotated in iTOL v6 [25]. The CXCN-6 isolate clustered closely with *Rhodotorula evergladensis* reference sequences (GenBank accession no. FJ008048.1), confirming its taxonomic identification as *R. evergladensis* CXCN-6.

### 2.4. Culture Medium and Extract Preparation

*R*. *evergladensis* was grown in the liquid medium (glucose 20 g/L, yeast extract 10 g/L, and peptone 20 g/L, pH 6.0). The growth of seed culture was monitored according to the optical density at 600 nm. When the seed culture reached the stationary phase, it was inoculated into 3 L of fermentation medium (same composition as seed culture: 20 g/L glucose, 10 g/L yeast extract, and 20 g/L peptone) in a 5 L fermenter at a 5% inoculation ratio. Fermentation was conducted for 96 h at 28 °C, 600 rpm, 1.0 vvm aeration, and pH 6.0. A high C/N ratio feeding solution (C/N = 105, g/g) was prepared using 500 g/L glucose, 5 g/L yeast extract, and 10 g/L peptone, and was supplied during the first 56 h of fermentation. After 16 h cultivation, feeding was started. The yeast cells were harvested every 24 h by centrifugation at 10,000 rpm for 10 min at 4 °C. The cells were washed twice with the same volume of sterile distilled water, then freeze-dried and weighted for the determination of dry cell weight (DCW).

The freeze-dried cells (100 mg) were hydrolyzed by 1.5 mL of 1 M HCl at 100 °C for 5 min, followed by centrifugation at 10,000 rpm for 3 min, then were washed with sterile distilled water. The hydrolyzed cells were gradually mixed with 1 mL acetone, then 0.5 mL ethyl acetate and 0.5 mL petroleum ether with bath ultrasonication shortly. Finally, the mixture was washed with 5 mL sterile distilled water, then centrifuged 4000 rpm 5 min, and the upper reddish color phase which contains carotenoids and other lipophilic compounds was collected; this extract process was repeated until a colorless pellet was obtained. The collected solvent mixture was dried in a rotary vacuum evaporator, then dissolved in 2 mL hexane [26]. The dissolved extract sample was centrifuged, filtrated through a 0.45 µm microfilter and stored at −80 °C for further analysis.

### 2.5. Carotenoid Analysis of R. evergladensis CXCN-6 Extract

The chromatographic analysis of the carotenoids present in the extract was performed in a reversed-phase High performance liquid chromatography (HPLC, Agilent 1260 series, Agilent Technologies, Santa Clara, CA, USA) with a DAD detector using a column (C18, 5 µm, 250 × 4.6 mm, Agilent, Cat# 880975-902). The mobile phases were acetonitrile and H_2_O (9:1, *v*/*v*) as eluent A, and pure ethyl acetate as eluent B. The gradient elution program was as follows: 0–5 min 100% A; 5–15 min 100% B; 15–30 min 100% B, according to the reported method with slight modification [26]. The column temperature was kept at 30 °C, and the injection volume was 20 µL. Chromatograms were recorded using the DAD detector. Although the absorbance maximum of torularhodin standard is at 500 nm, the torularhodin peak was higher and sharper at 490 nm; therefore, 490 nm was used for torularhodin and 450 nm for β-carotene. Both standards were prepared in ethyl acetate, and the CXCN-6 extract, initially dissolved in hexane, was evaporated and re-dissolved in ethyl acetate prior to HPLC analysis. The OpenLab CDS program was used for data processing. The compounds were identified on the basis of their retention time and UV–Vis spectra compared to torularhodin standard (Carotenature, Münsingen, Switzerland) and β-carotene (CFDA, Beijing, China, Cat# 100445). For HPLC analysis, both torularhodin and β-carotene standards were prepared in ethyl acetate. The CXCN-6 extract, initially dissolved in hexane, was evaporated and re-dissolved in ethyl acetate prior to injection. The content of torularhodin and β-carotene in the extract sample was calculated by standard curves generated using different concentrations of torularhodin and β-carotene standards.

Liquid chromatography–mass spectrometry (LC–MS) analysis was performed to further confirm the molecular identity of the carotenoids detected in the HPLC chromatograms. The analysis was conducted using an Agilent 6460 Triple Quadrupole LC–MS system (Agilent Technologies, USA) equipped with an atmospheric pressure chemical ionization (APCI) source operated in the positive ionization mode. Chromatographic separation was achieved on the same C18 column under identical gradient conditions as described for HPLC. The APCI parameters were optimized as follows: vaporizer temperature 400 °C, drying gas (N_2_) temperature 350 °C and flow rate 5 L min^−1^, corona discharge current 4 µA, nebulizer pressure 40 psi, and capillary voltage 3500 V. The MS data were acquired in full-scan mode within the mass range of *m*/*z* 300–700, followed by product ion scan for structural confirmation.

### 2.6. Lipid Analysis of R. evergladensis CXCN-6 Extract

The content of total lipids in the extract was calculated using the sulfo-phospho vanillin (SPV) method [27]. The phospho-vanillin reagent was freshly prepared by first dissolving 0.6 g of vanillin in 10 mL of absolute ethanol, followed by the addition of 90 mL of deionized water under continuous stirring. After complete mixing, 400 mL of 85% phosphoric acid was slowly added, and the final solution was stored protected from light until use. Briefly, 10 µL of extract sample was put into 20 mL glass bottle and dried at room temperature for 20 min to evaporate the solvent. Subsequently, 100 µL of distilled water was added to the extract sample, 2 mL of concentrated sulfuric acid was added to the sample slowly and was heated for 10 min at 100 °C and was cooled for 5 min in ice bath. 5 mL of freshly prepared phospho-vanillin reagent was then added, and the sample was incubated for 15 min at 37 °C incubator shaker at 200 rpm. The optical density (OD) was measured at 530 nm to quantify the lipid content within the sample. The calibration was carried out by treating different concentrations of olive oil with the same method as the sample. Fatty acids were quantified using a QTRAP 6500+ mass spectrometer (SCIEX, Framingham, MA, USA) coupled to an ACQUITY premier CSH (Waters, Milford, MA, USA). Chromatographic separation was performed on an ACQUITY premier CSH column (C18, 1.7 µm, 2.1 × 100 mm, Waters). Briefly, the solvent of 1 mL sample was evaporated at room temperature completely and reconstituted in 150 µL of eluent B. After vortexing and centrifuged, the supernatant was used for injection. The mobile phases consisted of (A) acetonitrile:water (6:4, *v*/*v*) with 10 mM ammonium formate and 0.1% formic acid; (B) isopropanol:acetonitrile (9:1, *v*/*v*) with 10 mM ammonium formate and 0.1% formic acid. The flow rate was set to 0.3 mL/min, and the injection volume was 2 µL. The gradient elution program was as follows: 0–2.5 min 95–70% A; 2.5–10.5 min 65% A; 10.5–12 min 50% A; 12–15 min 35% A; 15–18 min 1% A; 18–20 min 1% A; 20–20.5 min 1–95% A; 20.5–24 min 95% A. Mass spectrometric detection was carried out using QTRAP 6500+ ESI (−). Source parameters were as follows: source temperature, 550 °C; ion source gas 1, 55; gas 2, 50; curtain gas, 30; and ion spray voltage floating, 4500 V. Data acquisition was performed in scheduled multiple reaction monitoring (MRM) mode. The calibration was carried out by treating different concentrations of various fatty acid standards. Data acquisition and processing were performed using MultiQuant^TM^ 3.0.2 software (SCIEX).

### 2.7. Antioxidant Activity Assays

The antioxidant potential of *R. evergladensis* CXCN-6 extracts was evaluated using DPPH, ABTS radical scavenging assays, and an intracellular reactive oxygen species (ROS) assay.

#### 2.7.1. DPPH Radical Scavenging Activity

The DPPH radical scavenging activity was determined following the method of Blois (1958) with minor modifications [28]. Specifically, the CXCN-6 extract was dissolved in DMSO to prepare stock solutions, and all working concentrations (0.03–3.0 mg/mL) were obtained by diluting the stock in the corresponding assay buffer or culture medium. The final DMSO concentration did not exceed 0.1% (*v*/*v*) in any experiment. Briefly, 1 mL of 0.1 mM DPPH solution in ethanol was mixed with 1 mL of CXCN-6 extract at various concentrations (0.03–3.0 mg/mL). The reaction mixture was incubated in the dark at room temperature for 30 min, and absorbance was measured at 517 nm using a microplate reader (Thermo, Waltham, MA, USA). Vitamin C (1 mg/mL) was used as positive controls. The radical scavenging activity was expressed as the percentage decrease in absorbance compared with the control (without extract).

#### 2.7.2. ABTS Radical Cation Scavenging Activity

ABTS radical scavenging capacity was measured according to the method of Re et al. (1999) [29]. ABTS•^+^ radicals were generated by mixing 7 mM ABTS solution with 2.45 mM potassium persulfate and incubating the mixture in the dark for 16 h at room temperature. The resulting ABTS•^+^ solution was diluted with ethanol to obtain an absorbance of 0.70 ± 0.02 at 734 nm. Then, 1 mL of the ABTS•^+^ solution was mixed with 100 μL of CXCN-6 extract at different concentrations (0.3–3.0 mg/mL) and incubated for 6 min in the dark. Absorbance was measured at 734 nm, and scavenging activity was expressed as the percentage inhibition relative to the control. Vitamin C (1 mg/mL) was used as positive controls.

#### 2.7.3. Intracellular ROS Assay

Intracellular ROS scavenging activity was assessed using the DCF-DA fluorescent probe in UVA-irradiated HaCaT cells. Human keratinocytes (HaCaT) were cultured in DMEM supplemented with 10% FBS and 1% penicillin–streptomycin at 37 °C and 5% CO_2_. Cells were seeded into 96-well plates (1 × 10^4^ cells/well) and pre-treated with CXCN-6 extract (5–100 μg/mL) for 24 h. After washing with PBS, the cells were exposed to UVA irradiation (9 J/cm^2^) to induce oxidative stress, followed by incubation with 10 μM DCFH-DA for 30 min in the dark. Fluorescence intensity of the oxidized product DCF, representing intracellular ROS levels, was measured using a microplate reader (excitation at 485 nm and emission at 528 nm). In parallel, fluorescent images were captured using an inverted fluorescence microscope (Nikon Eclipse, Tokyo, Japan) to visualize intracellular ROS generation. The ROS inhibition rate was calculated relative to untreated and UVA-only control groups.

### 2.8. Anti-Inflammatory Activity Assay

The anti-inflammatory potential of *R. evergladensis* CXCN-6 extract was evaluated using a murine macrophage RAW264.7 cell model. Cells were cultured in Dulbecco’s Modified Eagle Medium (DMEM; Gibco, USA) supplemented with 10% fetal bovine serum and 1% penicillin–streptomycin at 37 °C in a humidified 5% CO_2_ incubator. For the assay, cells were seeded in 24-well plates at 2 × 10^5^ cells per well and pretreated with different concentrations of CXCN-6 extract (25, 50, and 100 μg/mL) for 2 h before stimulation with lipopolysaccharide (LPS, 1 μg/mL; Sigma-Aldrich, St. Louis, MO, USA) for 24 h. Vitamin C (50 μM) was used as a positive control. Following incubation, culture supernatants were collected for cytokine analysis. The levels of interleukin-6 (IL-6) and tumor necrosis factor-alpha (TNF-α) were determined using QuantiCyto^®^ ELISA kits (Proteintech, Wuhan, China) according to the manufacturer’s instructions, and absorbance was read at 450 nm using a microplate reader (Thermos, Singapore). Cell viability was assessed by the MTT assay to exclude cytotoxic effects, and all data were expressed as mean ± standard deviation (SD) from triplicate experiments. Statistical significance was evaluated using one-way ANOVA followed by Tukey’s test, with *p* < 0.05 considered significant.

### 2.9. Statistical Analysis

All experimental data were collected from at least three independent replicates and are presented as mean ± standard deviation. Statistical analyses were performed using GraphPad Prism version 9.5.1. Differences among groups were evaluated by one-way ANOVA followed by Tukey’s post hoc test, with *p* < 0.05 considered statistically significant.

## 3. Results

### 3.1. Isolation and Identification of R. evergladensis CXCN-6

A reddish-orange yeast strain, designated *R. evergladensis* CXCN-6, was isolated from the surface of the aquatic plant Nymphaea “Gorgeous Purple,” collected from Kunming, Yunnan Province, China—a region characterized by strong ultraviolet (UV) radiation and high solar intensity (Figure 1). The sampling environment was selected due to its potential to harbor microorganisms adapted to oxidative stress, which may exhibit enhanced carotenoid biosynthesis. Primary isolation was performed on YPD plate after enrichment in liquid YPD medium at 28 °C for 24 h with gentle shaking (150 rpm). After streaking onto YPD plates and incubation at 28 °C for 48 h, colonies of CXCN-6 appeared circular, convex, smooth, and glossy, displaying a distinctive reddish-orange pigmentation indicative of carotenoid accumulation (Figure 1). The isolate was purified through three successive streaking steps under identical conditions to ensure genetic and phenotypic stability.

For molecular identification of CXCN-6, genomic DNA was extracted. The internal transcribed spacer (ITS) region of ribosomal DNA was amplified using the universal primers ITS1 (5′-TCCGTAGGTGAACCTGCGG-3′) and ITS4 (5′-TCCTCCGCTTATTGATATGC-3′). PCR products (542 bp) were purified and sequenced. BLASTn on the NCBI online server (https://blast.ncbi.nlm.nih.gov) analysis of the ITS sequence revealed ≥ 99% identity with *R. evergladensis* reference sequences in GenBank (accessions: NR_137709.1). A phylogenetic tree constructed using the neighbor-joining method confirmed that strain CXCN-6 clustered tightly with *R. evergladensis* type strains and was distinct from other *Rhodotorula* species, including *R. mucilaginosa*, *R. glutinis*, and *R. minuta* (Figure 2). This confirmed the strain’s identity as *R. evergladensis*. To our knowledge, *R. evergladensis* remains a relatively uncharacterized yeast species, with scarce reports addressing its metabolic or functional potential. The strain has been deposited in the China General Microbiological Culture Collection Center (CGMCC) under the accession number CGMCC 7.610 for future reference and industrial exploration.

### 3.2. Whole-Genome Sequencing and Functional Annotation of R. evergladensis CXCN-6

To comprehensively understand the molecular determinants underlying the unique metabolic plasticity of *R. evergladensis* CXCN-6, whole-genome sequencing and functional annotation were performed. Genomic characterization provides fundamental insights into the genetic determinants that enable this strain to simultaneously produce carotenoids and lipids—two biosynthetically demanding secondary metabolites linked to stress adaptation and cellular redox balance [30]. Understanding its genome organization thus establishes a foundation for exploring metabolic regulation and guiding future strain improvement for industrial applications.

High-quality genomic DNA was sequenced using a hybrid Illumina–Oxford Nanopore platform, ensuring both high accuracy and long-read continuity. The assembled genome comprised approximately 20.72 Mb with a GC content of 62.17%, distributed across 18 scaffolds with an N50 of 1.4 Mb (Appendix A). BUSCO analysis indicated ~90% completeness, confirming high assembly quality suitable for downstream annotation. A total of 6389 protein-coding genes and 105 tRNA genes were predicted (Appendix A). Functional annotation against KEGG, GO, and COG databases revealed that *R. evergladensis* CXCN-6 is enriched in genes related to central carbon metabolism, fatty acid biosynthesis, and carotenoid formation, suggesting a well-organized metabolic framework supporting dual metabolite synthesis.

Key genes of the mevalonate (MVA) pathway—such as HMG-CoA reductase, farnesyl pyrophosphate synthase, and geranylgeranyl pyrophosphate synthase—were identified, providing the precursors for carotenoid biosynthesis. The downstream carotenoid pathway was complete, including *crtB* (phytoene synthase), *crtI* (phytoene desaturase), *crtY* (lycopene cyclase), and *crtA* (carotenoid oxygenase), consistent with the observed torularhodin production (Figure 3). Genes involved in lipid metabolism, including FAS1/FAS2 (fatty acid synthase), ACC1 (acetyl-CoA carboxylase), and multiple desaturases (Δ9, Δ12, Δ15), were also identified, explaining the accumulation of long-chain unsaturated fatty acids (Figure 3). Moreover, the presence of oxidative stress–related genes, such as superoxide dismutase, catalase, and glutathione peroxidase, supports the coordination between antioxidant defense and carotenoid synthesis. Collectively, these genomic features demonstrate that *R. evergladensis* CXCN-6 harbors a complete and well-integrated genetic system for the co-production of carotenoids and lipids, highlighting its potential as a robust and engineerable yeast platform for sustainable biomanufacturing.

### 3.3. Torularhodin Production and Spectral Characterization in R. evergladensis CXCN-6

The *R. evergladensis* CXCN-6 strain exhibited an intense reddish-orange pigmentation in both liquid and solid cultures (Appendix A), consistent with the characteristic appearance of carotenoid-producing *Rhodotorula* species. The intensity of pigmentation increased progressively over the course of cultivation, indicating substantial intracellular pigment accumulation as biomass increased. Although this observation is qualitative, it provides an initial indication of active carotenoid production and supports the need for detailed analytical quantification. CXCN-6 extracts of the intracellular pigments showed a strong absorption maximum at 490 nm (Appendix A). These spectral features are typical of polyene carotenoids with extended conjugated double-bond systems, confirming the presence of chromophoric molecules such as torularhodin, torulene, and β-carotene. The steady increase in absorption intensity correlated with cell dry weight during culture progression, indicating proportional intracellular accumulation of carotenoids with biomass growth.

High-performance liquid chromatography (HPLC) analysis using a C18 reverse-phase column (detection at 490 nm) revealed a dominant pigment peak with a retention time identical to the torularhodin standard (Figure 4A). Two minor peaks, detected at retention times of 18.6 min and 19.5 min, corresponded to β-carotene and torulene, respectively, suggesting their roles as biosynthetic intermediates within the carotenoid pathway (Figure 4A). Quantitative integration of chromatographic areas indicated that torularhodin accounted for more than 90% of the total carotenoid content, establishing it as the predominant carotenoid pigment synthesized by *R. evergladensis* CXCN-6. Further structural verification was achieved through liquid chromatography–mass spectrometry (LC–MS) analysis operated in atmospheric pressure chemical ionization (APCI^+^) mode. The predominant pigment exhibited a molecular ion peak at *m*/*z* 563.4 [M]^+^ (Figure 4B), corresponding to the theoretical molecular weight of torularhodin (C_40_H_56_O_2_, MW = 564.4). This spectral pattern was consistent with previously reported data for torularhodin isolated from *Rhodotorula* sp., thereby confirming its structural identity [21].

### 3.4. Fatty Acid Profile of R. evergladensis CXCN-6

To further elucidate the metabolic potential of *R. evergladensis* CXCN-6, lipid composition was analyzed. Total lipids were extracted for LC–MS analysis [31]. Chromatographic separation on a reverse-phase C18 column revealed that the lipid fraction was dominated by C18-series fatty acids. The major unsaturated components were linoleic acid (C18:2, 46.6%), oleic acid (C18:1, 22.8%), and α-linolenic acid (C18:3, 19.5%), while palmitic acid (C16:0, 10.0%) represented the main saturated species (Figure 5A). This fatty acid profile—rich in polyunsaturated long-chain fatty acids (PUFAs, ~90% of total fatty acids)—is characteristic of stress-tolerant *Rhodotorula* species and indicates enhanced membrane fluidity and oxidative resilience. The simultaneous accumulation of PUFA-enriched lipids and the potent antioxidant torularhodin underscores the strain’s metabolic synergy and environmental adaptability. These results highlight *R. evergladensis* CXCN-6 as a metabolically versatile and robust oleaginous yeast suitable for producing nutraceutical, cosmetic, and functional food ingredients.

### 3.5. Fermentation Performance of R. evergladensis CXCN-6

To evaluate the production performance of *R. evergladensis* CXCN-6, fermentation was first examined under batch conditions, followed by a fed-batch strategy. In the initial batch culture (no feeding), the medium composition was identical to the seed culture (20 g/L glucose, 10 g/L yeast extract, 20 g/L peptone). The growth profile showed that biomass accumulation slowed markedly after approximately 72 h, indicating the transition from exponential growth to the stationary phase (Appendix A). During this period, pigment coloration intensified, suggesting that secondary metabolite accumulation occurred primarily after active biomass formation.

Based on the batch experiment, a fed-batch fermentation was subsequently performed in a 5 L bioreactor using the same basal medium (3 L working volume). Process conditions were maintained at 28 °C, 600 rpm, 1 vvm aeration, and pH 6.0. A high C/N-ratio feeding solution (500 g/L glucose, 5 g/L yeast extract, 10 g/L peptone; C/N = 105 g/g) was supplied during early fermentation to sustain biomass growth. Feeding was intentionally stopped at 54 h to impose mild nutrient limitation, a strategy commonly used to trigger carotenoid and lipid accumulation in oleaginous yeasts (Appendix A). The culture reached the stationary phase at ~72 h with a final dry cell weight (DCW) of 50 g/L.

Under this fed-batch strategy, *R. evergladensis* CXCN-6 produced 63.56 mg/L of torularhodin and 9.83 g/L of total lipids (Figure 5B,C). On a cell dry weight basis, torularhodin and total lipids reached 1.27 mg/g DCW and 142.6 mg/g DCW, respectively, representing the highest intracellular contents during the fermentation process (Appendix A). These data demonstrate that nitrogen-limited and oxidative conditions effectively redirected cellular metabolism toward secondary product formation, maximizing the accumulation of both torularhodin and storage lipids.

Compared with the unoptimized shake-flask culture, the final lipid concentration increased from 0.73 g/L to 9.83 g/L (approximately a 13-fold improvement), while the final torularhodin concentration increased from 15.3 mg/L to 63.56 mg/L (a 4.2-fold enhancement) (Table 1). Among the carotenoid components, torularhodin became the predominant pigment, accounting for more than 95% of total carotenoids in the optimized culture, compared with approximately 90% before optimization. Notably, the β-carotene content per dry cell weight (DCW) slightly declined (from 90.0 μg/g DCW to 58.9 μg/g DCW), whereas torularhodin content significantly increased (from 0.77 mg/L to 1.27 mg/L). This inverse trend indicates a metabolic flux shift toward the oxidative branch of the carotenoid biosynthetic pathway, favoring the conversion of β-carotene and torulene into torularhodin.

Such a shift is likely driven by enhanced activity of carotenoid oxygenases and desaturases under nitrogen-limited and oxidative stress conditions, where reactive oxygen species (ROS) accumulation acts as a signal to promote carotenoid oxidation. The resulting enrichment of torularhodin not only strengthens the antioxidant capacity of the culture but also reflects a refined stress adaptation mechanism characteristic of high-performance *Rhodotorula* species. These findings confirm that *R. evergladensis* CXCN-6 possesses strong metabolic flexibility and productivity, supporting its potential as a robust platform for industrial-scale bioproduction of natural antioxidants.

### 3.6. Stability of R. evergladensis CXCN-6 Extracts

To assess the physicochemical stability of the *R. evergladensis* CXCN-6 extract, particularly the robustness of torularhodin, the CXCN-6 extract was stored under different temperature and light exposure conditions. Samples were incubated at room temperature (25 °C), 37 °C, and 45 °C for over three months under both light-protected and light-exposed conditions. Remarkably, no visible color fading or pigment degradation was observed throughout the storage period (Appendix A). To quantify these changes, absorbance at 490 nm was monitored over a 16-week storage period. At 25 °C, the extract maintained over 95% of its initial absorbance, demonstrating excellent stability with minimal degradation. At 37 °C, pigment retention remained above 80%, indicating only moderate loss. A more pronounced decline was observed at 45 °C, where approximately 40% of pigment intensity was lost by week 16, though substantial coloration was still retained (Appendix A). Importantly, light-exposed and dark-stored samples showed nearly identical degradation patterns at all temperatures, confirming that thermal stress—rather than photochemical processes—is the primary factor influencing pigment stability in the CXCN-6 extract (Appendix A).

The comparable results between light-exposed and dark-stored samples suggest that torularhodin possesses intrinsic photostability and thermal resistance, likely attributable to its extended conjugated double-bond system and terminal carboxyl functional group, which enhance molecular rigidity and oxidative resilience. This exceptional stability distinguishes torularhodin from less oxygenated carotenoids such as β-carotene, which are typically prone to photooxidation and thermal degradation. The stable coloration and spectral integrity of the CXCN-6 extract highlight its suitability for applications requiring color and oxidative stability—such as in nutraceutical, cosmetic, and food formulations.

### 3.7. Antioxidant Activity of R. evergladensis CXCN-6 Extracts

The antioxidant potential of the CXCN-6 extract from was comprehensively evaluated using chemical radical scavenging assays (DPPH and ABTS) and a cell-based intracellular reactive oxygen species (ROS) assay (Figure 6). Both chemical assays showed a distinct concentration-dependent response, indicating the presence of active metabolites capable of neutralizing free radicals. In the DPPH assay, the extract displayed progressive scavenging activity, reaching approximately 50% inhibition at a concentration of 3 mg/mL (Figure 6A). Similarly, in the ABTS^+^· assay, stronger scavenging capacity was observed, with inhibition levels approaching 90% at the same concentration (Figure 6B).

The strong antioxidant capacity of the CXCN-6 extract may be partially attributed to the presence of torularhodin and unsaturated lipid components, which are known to possess intrinsic redox-balancing properties. Torularhodin, as an oxygenated carotenoid with an extended conjugated double-bond system, has been reported to effectively scavenge reactive oxygen species [32], while PUFAs could enhance membrane-associated oxidative defense [33]. These findings suggest that *R. evergladensis* CXCN-6 has evolved an effective biochemical defense strategy against oxidative stress, likely reflecting its adaptation to high-UV, oxygen-rich environments. Collectively, the results support CXCN-6 extract as a promising natural antioxidant resource.

To further confirm its cellular antioxidant performance, UVA-induced oxidative stress experiments were conducted in RAW264.7 macrophages using a 9 J/cm^2^ UVA irradiation model. UVA exposure markedly increased intracellular ROS levels, as detected by the DCFH-DA fluorescent probe, indicating oxidative stress. Pre-treatment with CXCN-6 extract significantly reduced intracellular fluorescence intensity in a dose-dependent manner, suggesting effective scavenging of ROS and protection against UVA-induced oxidative injury. At 100 µg/mL, intracellular ROS generation was reduced to less than 40% of the UVA-only control, confirming potent cellular antioxidant activity (Figure 6C, D).

These results collectively demonstrate that the *R. evergladensis* CXCN-6 extract possesses both chemical and cellular antioxidant capabilities. The strong scavenging effect is primarily attributed to torularhodin, a carotenoid with high singlet oxygen quenching efficiency, and polyunsaturated fatty acids (PUFAs), which contribute to redox homeostasis and membrane stability. To contextualize the antioxidant performance of the CXCN-6 extract, we compared our findings with previously reported activities of common carotenoids. Published studies have shown that torularhodin generally exhibits stronger radical-scavenging activity than several widely studied carotenoids [34,35]. For example, torularhodin has been reported to display higher peroxyl radical–quenching efficiency than β-carotene and lycopene in chemical assays, and to show ORAC values comparable to or slightly exceeding those of astaxanthin. These reference datasets indicate that torularhodin belongs to the group of high-efficiency antioxidant carotenoids. The synergistic action of these bioactive components likely underlies the yeast’s intrinsic oxidative stress resistance and supports its potential application in nutraceutical, cosmeceutical, and photoprotective formulations.

### 3.8. Anti-Inflammatory Activity of R. evergladensis CXCN-6 Extracts

The anti-inflammatory effects of the CXCN-6 extract were subsequently examined in LPS-stimulated RAW264.7 macrophages, a widely used model for evaluating inflammatory responses [36]. Stimulation with LPS alone induced a pronounced upregulation of pro-inflammatory cytokines, particularly interleukin-6 (IL-6) and tumor necrosis factor-alpha (TNF-α), confirming macrophage activation and inflammatory signaling (Figure 7). Co-treatment with the extract of CXCN-6 markedly and dose-dependently suppressed cytokine secretion (Figure 7). At 100 µg/mL, IL-6 production decreased by more than 90%, while TNF-α levels were reduced by approximately 40% compared to the LPS-only control group. Importantly, cell viability assays confirmed that the extract exhibited no cytotoxic effects at all tested concentrations, indicating that the observed inhibition of cytokine release resulted from genuine anti-inflammatory action rather than cellular toxicity.

Given the strong antioxidant activity of the extract demonstrated in the UVA-induced oxidative stress model, it is plausible that the reduction in inflammatory cytokines is associated with the attenuation of oxidative stress-mediated signaling pathways. Carotenoids such as torularhodin, known for their potent singlet oxygen and free radical scavenging properties, likely play a central role in modulating intracellular redox balance. Additionally, the presence of PUFAs in the extract may further contribute to anti-inflammatory efficacy by maintaining membrane fluidity and interfering with eicosanoid biosynthesis. Collectively, these results demonstrate that *R. evergladensis* CXCN-6 extract exerts dual antioxidant and anti-inflammatory activities at the cellular level. The coordinated suppression of oxidative stress and inflammatory cytokines highlights its potential as a valuable natural source for nutraceutical, cosmeceutical, and functional food applications, particularly in formulations aimed at mitigating photooxidative or inflammatory skin damage.

## 4. Discussion

This study provides the first functional characterization of *R*. *evergladensis* as a carotenoid- and lipid-producing yeast. The results demonstrate its strong capacity to synthesize torularhodin, a distinctive oxygenated carotenoid featuring an extended conjugated polyene chain and a terminal carboxyl group. Compared with β-carotene and torulene, torularhodin exhibits superior antioxidant activity and oxidative stability, enabling more effective neutralization of singlet oxygen and free radicals [6,37,38]. Its dominance in CXCN-6 implies an active oxidative branch of the carotenoid biosynthetic pathway, promoting torulene oxidation to torularhodin as part of an adaptive defense mechanism. The genomic detection of carotenoid oxygenase and aldehyde dehydrogenase homologs further supports the presence of this oxygenation module, consistent with previous findings in *Rhodotorula glutinis* and *R. mucilaginosa* [7,39,40].

In parallel, CXCN-6 accumulated significant amounts of PUFAs—including linoleic (C18:2), oleic (C18:1), and α-linolenic (C18:3) acids—indicative of a highly active desaturation network. Such lipid composition suggests enhanced membrane fluidity, oxidative tolerance, and adaptation to fluctuating environmental stress, properties that are advantageous for industrial biocatalysts [41]. The coexistence of unsaturated lipids and carotenoids reflects coordinated intracellular regulation rather than functional synergy in the extract. In living cells, lipid droplets can act as reservoirs for hydrophobic pigments, and previous studies suggest possible interactions between carotenoids and unsaturated fatty acids under hypoxic cellular conditions that support redox balance [42,43]. This co-regulation likely arises from shared precursors such as acetyl-CoA and NADPH and is typically triggered under nutrient limitation or oxidative stress [26,44]. Importantly, these intracellular relationships should not be extrapolated to extract behavior: highly unsaturated fatty acids may oxidize readily when exposed to air and therefore do not enhance carotenoid stability in vitro. The antioxidant effects measured in this study thus primarily reflect the intrinsic chemical properties of torularhodin.

Recent studies have shown that many oleaginous red yeasts—including *Rhodotorula* and *Rhodosporidium* species—co-produce carotenoids and storage lipids in response to nutrient limitation or environmental stress [45,46]. For instance, *Rhodosporidium toruloides* can synthesize both lipids and oxygenated carotenoids such as torularhodin, and plasma-mutagenized strains have achieved torularhodin levels of 481.92 μg/g DCW. Likewise, *R. glutinis*, *R. mucilaginosa*, and *Rhodotorula gracilis* have been shown to increase both lipid content and torularhodin under salinity, oxidative stimuli, or temperature shifts. Our observations in CXCN-6 follow this established pattern: both lipids and carotenoids increased markedly after feeding cessation and the introduction of mild starvation, indicating that CXCN-6 responds to nutrient limitation similarly to other high-performing red yeasts. These results expand the number of red-yeast species with demonstrated dual-production capabilities and reinforce the value of CXCN-6 for biotechnological development.

The antioxidant and anti-inflammatory activities observed in CXCN-6 extracts also correspond well with previous reports describing the strong bioactivity of torularhodin and related carotenoids. Torularhodin has been documented to possess higher radical-scavenging capacity than β-carotene or lycopene in chemical assays, and torularhodin-rich extracts from *Rhodotorula* species have demonstrated the ability to reduce intracellular ROS and modulate cytokine responses in cell models. Although the CXCN-6 extract is a complex mixture containing both carotenoids and lipids, its measured antioxidant and anti-inflammatory properties fall within the ranges reported for torularhodin-enriched preparations. This suggests that the bioactivities detected here are broadly comparable to established findings, while still reflecting the unique compositional profile of CXCN-6.

The combined presence of torularhodin and PUFA-rich lipids constitutes a comprehensive antioxidative strategy. Torularhodin scavenges reactive oxygen species (ROS) and modulates cellular defense signaling through the inhibition of NF-κB activation, leading to a reduction in pro-inflammatory cytokine production [35,47,48]. At the same time, the accumulation of PUFAs enhances membrane elasticity, regulates permeability, and provides resilience against lipid peroxidation under oxidative stress [49,50]. This dual system, involving both enzymatic and non-enzymatic defense mechanisms, likely evolved as an adaptive response to the yeast’s native environment—the exposed, high-UV, and nutrient-variable surface of Nymphaea “Gorgeous Purple” leaves in the high-altitude plateau region of Yunnan. The potent antioxidant and anti-inflammatory properties of CXCN-6 extract thus highlight its potential as a valuable bioactive ingredient for health-related industries.

The carotenoid biosynthetic capacity of *R. evergladensis* CXCN-6 was notably superior to that of previously reported *Rhodotorula* species (Table 2). Under optimized fermentation conditions, CXCN-6 produced up to 63.6 mg/L of torularhodin, which accounted for more than 95% of total carotenoids. This yield is markedly higher than those documented for other *Rhodotorula* yeasts, including *R. glutinis*, *R. toruloides*, and *R. rubra*, which typically produce 1.4–35.6 mg/L under similar cultivation conditions [43,51,52,53,54,55]. The predominance of torularhodin suggests that CXCN-6 possesses a highly active oxidative branch within the carotenoid biosynthetic pathway, enabling efficient conversion from torulene to torularhodin. Such enhanced oxidation capacity may be a result of environmental adaptation, as the strain was isolated from the UV-intense surface of Nymphaea “Gorgeous Purple” leaves in Yunnan. The high-level production of torularhodin therefore reflects both genetic and ecological optimization, establishing *R. evergladensis* CXCN-6 as one of the most efficient natural torularhodin producers reported to date and a promising microbial platform for industrial carotenoid biosynthesis.

From a biotechnological perspective, the co-production of torularhodin and PUFA-rich lipids within a single microbial chassis offers significant process and economic benefits. The integration of pigment and lipid biosynthesis enables shared use of key precursors such as acetyl-CoA and glycerol-3-phosphate, improving carbon flux efficiency and overall metabolic economy. This intrinsic coupling enhances carbon conversion and streamlines downstream processing, as both metabolites can be co-extracted from one biomass source. Such integration is particularly attractive for large-scale fermentation, where reduced solvent use, simplified extraction, and minimized waste directly lower production costs and improve sustainability. Furthermore, embedding torularhodin within endogenous lipids enhances pigment stability, oxidative protection, and bioavailability—properties valuable for nutraceutical and cosmetic formulations. Together, these features highlight *R. evergladensis* CXCN-6 as a promising microbial platform for green biomanufacturing.

Future work should emphasize the genetic and metabolic elucidation of torularhodin biosynthesis through integrative multi-omics approaches—combining genomics, transcriptomics, metabolomics, and fluxomics—to identify rate-limiting steps and key regulatory nodes linking carotenoid and lipid pathways. Engineering strategies that boost precursor supply, such as improving acetyl-CoA flux, NADPH regeneration, and oxygenase activity, could further enhance de novo synthesis of torularhodin and microbial oils [56,57]. In parallel, CRISPR-based editing, adaptive evolution, and dynamic regulatory tools may optimize redox balance, stress tolerance, and substrate utilization. Industrially, the use of renewable carbon sources such as lignocellulosic hydrolysates, agricultural residues, or waste glycerol can reduce costs while promoting biowaste valorization and circular bioeconomy development [58,59]. Through these combined advances in systems biology and process engineering, *R. evergladensis* CXCN-6 holds strong potential for scalable, eco-efficient production of high-value carotenoids and functional lipids.

## 5. Conclusions

In conclusion, *R. evergladensis* CXCN-6 represents a metabolically versatile yeast capable of simultaneously producing the potent antioxidant torularhodin and PUFA-enriched lipids. Its genomic features, physiological adaptability, and optimized fermentation performance collectively underscore its value as a promising microbial platform for natural pigment and lipid co-production. Through integrated process optimization, metabolic pathway elucidation, and the strategic use of renewable substrates, CXCN-6 could support eco-friendly and scalable production of carotenoids and functional oils. Beyond its industrial relevance, the strain also provides a valuable model for studying oxidative stress adaptation and secondary metabolism in non-conventional yeasts. Continued exploration of its regulatory networks and bioprocess scalability will further enhance its potential applications in nutraceutical, cosmetic, and functional food industries, advancing sustainable biomanufacturing and resource-efficient production of natural bioactives.

## Figures and Tables

**Figure 1 antioxidants-14-01420-f001:**
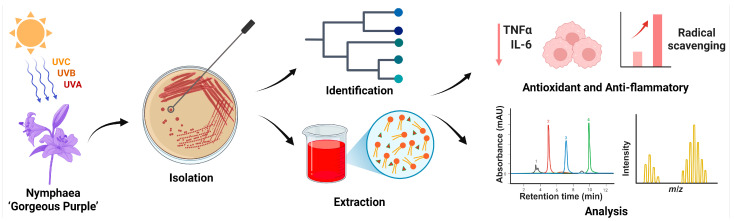
Workflow of isolation, characterization, and bioactivity evaluation of *R. evergladensis* CXCN-6. This schematic summary illustrates the experimental workflow of the study. (1) A pigmented yeast strain, *R. evergladensis* CXCN-6, was isolated from the surface of Nymphaea “Gorgeous Purple” leaves exposed to intense ultraviolet (UV) radiation in plateau environment. (2) The strain was taxonomically identified based on ITS rRNA gene sequencing and phylogenetic analysis. (3) Intracellular bioactive compounds, including the carotenoid torularhodin and storage lipids, were extracted from cultured biomass. (4) The chemical composition and structure of the metabolites were characterized using HPLC and LC–MS analyses. (5) The antioxidant and anti-inflammatory activities of the CXCN-6 extracts were subsequently evaluated through biochemical assays and cell-based functional experiments, confirming their potent ROS-scavenging and cytoprotective effects.

**Figure 2 antioxidants-14-01420-f002:**
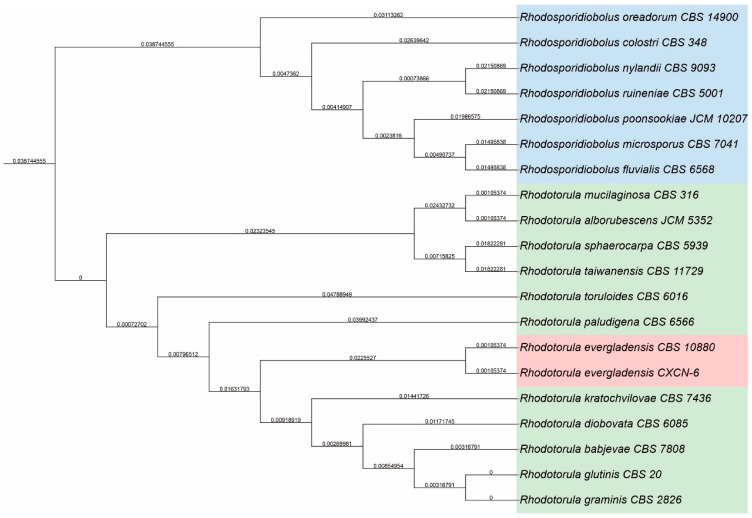
Phylogenetic analysis of *R. evergladensis* CXCN-6 based on ITS rDNA sequences. The phylogenetic tree was constructed using the neighbor-joining method with 1000 bootstrap replications in MEGA 7. The ITS sequence of *R. evergladensis* CXCN-6 was aligned with representative sequences of closely related *Rhodotorula* species retrieved from the NCBI GenBank database. Bootstrap values (%) are indicated at the nodes. Cryptococcus neoformans was used as the outgroup to root the tree. The clustering pattern clearly places strain CXCN-6 within the *R. evergladensis* clade, confirming its taxonomic identity.

**Figure 3 antioxidants-14-01420-f003:**
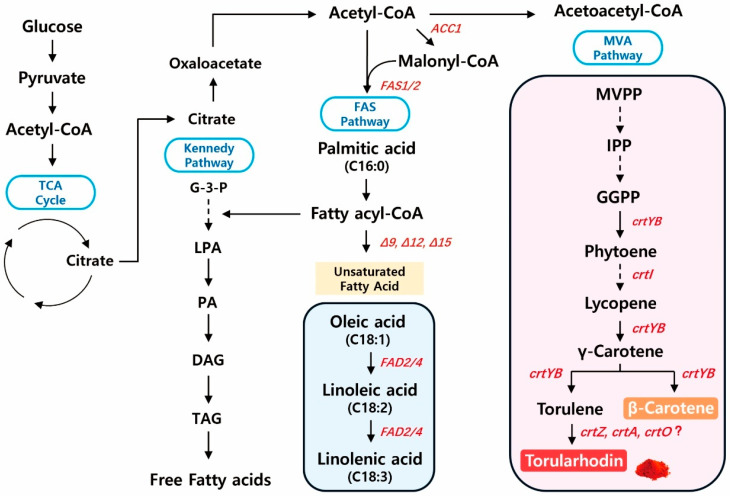
Overview of metabolic pathways for torularhodin and unsaturated fatty acid biosynthesis in *R. evergladensis* CXCN-6. This schematic illustrates the central carbon metabolism and its branching into carotenoid and lipid biosynthetic routes. Glucose enters the glycolytic (EMP) and tricarboxylic acid (TCA) pathways to generate acetyl-CoA, a key metabolic precursor. Acetyl-CoA serves as the starting substrate for both the mevalonate pathway—leading to isoprenoid intermediates (IPP, FPP, GGPP) and downstream carotenoids including phytoene, lycopene, β-carotene, torulene, and torularhodin—and the fatty acid synthesis pathway generating long-chain and polyunsaturated fatty acids (PUFAs). Enzymes and intermediates involved in the metabolic pathways are highlighted in red. The interconnection between torularhodin and PUFA production reflects shared carbon flux and redox cofactor balance, forming the biochemical basis for co-production of pigments and lipids in *R. evergladensis* CXCN-6.

**Figure 4 antioxidants-14-01420-f004:**
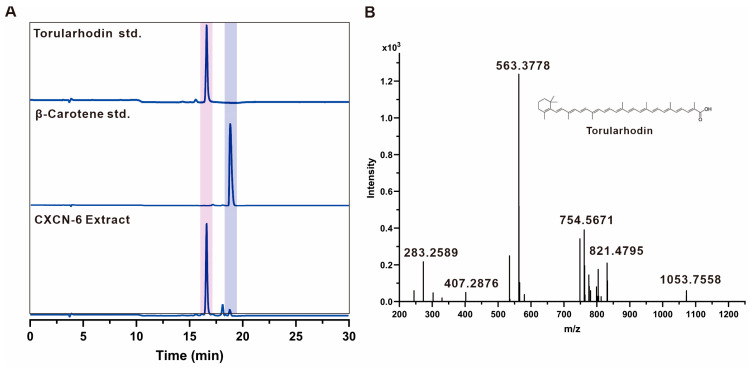
Carotenoid profile of *R. evergladensis* CXCN-6 extract. (**A**) HPLC chromatogram (C18 column, 490 nm) showing the major carotenoid peak corresponding to torularhodin. (**B**) LC–MS analysis (APCI^+^) confirming torularhodin as the dominant carotenoid based on its characteristic molecular ion peak (*m*/*z* 563.4).

**Figure 5 antioxidants-14-01420-f005:**
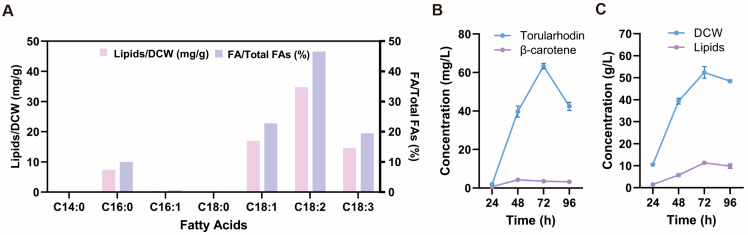
Lipid profile and fermentation performance of *R. evergladensis* CXCN-6 extract. (**A**) LC–MS chromatogram of fatty acid (FA) methyl esters, indicating linoleic (C18:2), oleic (C18:1), α-linolenic (C18:3), and palmitic acid (C16:0) as major components. (**B**) Time-course production of torularhodin and β-carotene during 5 L fermentation. (**C**) Time-course of dry cell weight (DCW) and total lipid accumulation.

**Figure 6 antioxidants-14-01420-f006:**
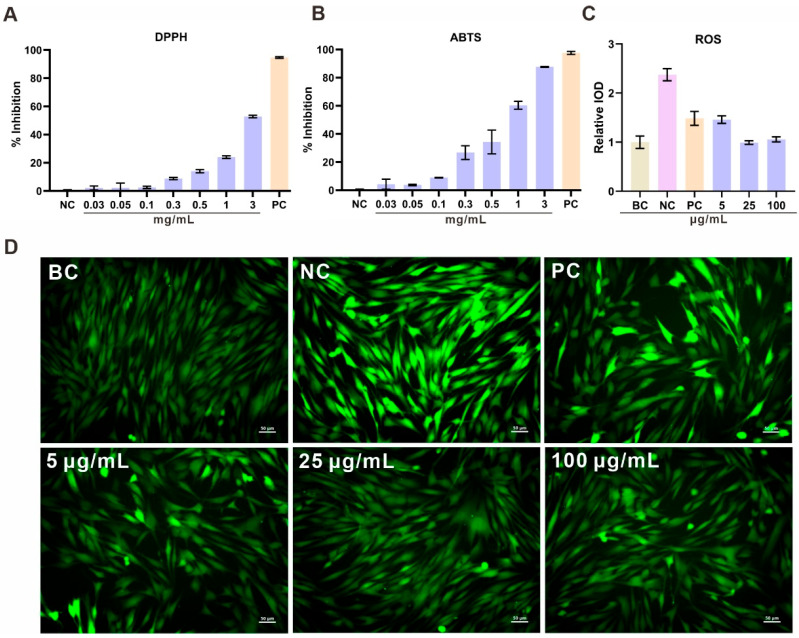
Antioxidant Activity of *R. evergladensis* CXCN-6 Extracts. (**A**) DPPH radical scavenging activity of CXCN-6 extract at concentrations ranging from 0.03 to 3 mg/mL. The scavenging rate increased in a dose-dependent manner, reaching approximately 50% at 3 mg/mL. (**B**) ABTS^+^· radical scavenging activity of the extract under the same concentration range, showing a stronger response with ~90% inhibition at 3 mg/mL. (**C**) Intracellular ROS quantification in UVA-irradiated HaCaT cells (9 J/cm^2^). Cells pretreated with CXCN-6 extract (5–100 μg/mL) displayed significant, concentration-dependent ROS suppression relative to the UVA-only group (n = 3, *p* < 0.05). Vitamin C (100 μg/mL) and Vitamin E (7 μg/mL) mixture was used as a positive control. (**D**) Representative fluorescence microscopy images of DCF-DA-stained HaCaT cells. The intense green fluorescence induced by UVA exposure was markedly reduced following extract treatment, indicating intracellular ROS scavenging. Scale bar = 50 μm. NC: Negative Control; PC: Positive Control; BC: Blank Control.

**Figure 7 antioxidants-14-01420-f007:**
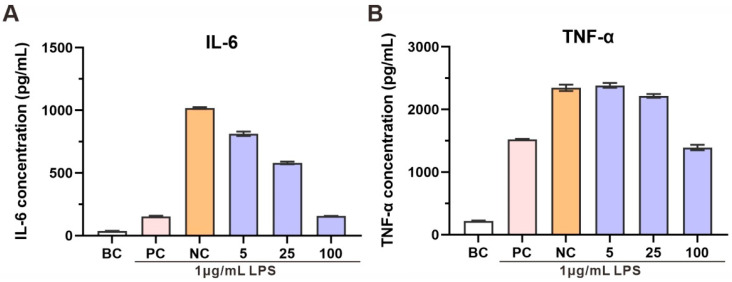
Anti-inflammatory Activity of *R. evergladensis* CXCN-6 Extracts. (**A**) Effect of CXCN-6 extract on interleukin-6 (IL-6) secretion in LPS-stimulated RAW 264.7 macrophages. LPS treatment markedly increased IL-6 release, whereas co-treatment with the extract (25–100 µg/mL) significantly suppressed cytokine production in a concentration-dependent manner (*p* < 0.05). (**B**) Effect of CXCN-6 extract on tumor necrosis factor-α (TNF-α) secretion under the same conditions. The extract reduced TNF-α levels by approximately 40% at 100 µg/mL. Vitamin C was used as a positive control. Data are presented as mean ± SD (n = 3); asterisks indicate significant differences compared with the LPS group (*p* < 0.05). NC: Negative Control; PC: Positive Control; BC: Blank Control.

**Table 1 antioxidants-14-01420-t001:** Comparison of lipid and carotenoid production in *R. evergladensis* CXCN-6 before and after fermentation optimization.

CXCN-6 Extract	Production (mg/L)		Production/DCW (mg/g)	
	Before Optimization	After Optimization	Before Optimization	After Optimization
DCW	20	50	/	/
Total lipids	733.3	9830.0	36.7	196.6
Torularhodin	15.3	63.56	0.77	1.27
β-Carotene	1.80	2.90	0.090	0.058

**Table 2 antioxidants-14-01420-t002:** Comparison of torularhodin production by *R. evergladensis* CXCN-6 with other *Rhodotorula* species under various culture conditions.

Yeast Strain	Carbon Source	Nitrogen Source	Torularhodin	β-Carotene	Reference
*R. evergladensis* CXCN-6	Glucose	Yeast extract, Peptone	63.6 mg/L	2.9 mg/L	This study
*R. rubra* PTCC 5255	Glucose	Ammonium sulfate	35.6 mg/L	1.0 mg/L	[51]
*R. glutinis* JMT 21978	Glucose	Yeast extract	6.6 mg/L	2.0 mg/L	[43]
*R. glutinis* ZHK	Dextrose	Yeast extract, Peptone	1.4 mg/L	1.7 mg/L	[52]
*R. toruloides* A1-15-BRQ	Glucose	Ammonium sulfate	21.3 mg/L	1.2 mg/L	[53]
*R. toruloides* CBS 5490	Glycerol	Yeast extract, Peptone	19.7 mg/L	6.8 mg/L	[54]
*R. toruloides* M18	Glucose	Yeast extract, Peptone	8.95 mg/L	285.5 mg/L	[55]

## Data Availability

The data supporting the findings of this study are available from the corresponding author upon reasonable request. All other relevant experimental data, including raw measurements, chromatographic profiles, spectral data, and microscopy images, are provided within the main manuscript and its Appendix A. Additional datasets generated or analyzed during the current study, such as HPLC chromatograms, LC–MS spectra, and fermentation monitoring records, can be made available by the authors upon justified request.

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
