# Peer review of "A Novel *Rhodotorula evergladensis* CXCN-6 Rich in Torularhodin and PUFAs with Potent Antioxidant and Anti-Inflammatory Activities"

_antioxidants, 2025, doi:10.3390/antiox14121420_

Round 1
Reviewer 1 Report
In this manuscript, the authors are analyzing the carotenoid and unsaturated fatty acid production characteristics of Rhodotorula evergladensis CXCN-6. Furthermore, They are also evaluating its functionality, such as antioxidant and anti-inflammatory properties. This research is interesting in terms of the use of microorganisms in food and processed foods. It is also interesting that it accumulates the unique carotenoid torularhodin. There are some requests for author.
1. 3.7 Antioxidant activity of R. evergladensis CXCN-6 Extracts
↓
The authors used vitamin C and vitamin E as positive controls. I am interested in the unique carotenoid torularhodin contained in R. evergladensis CXCN-6. So, a comparison with other common carotenoids would be beneficial. Alternatively, it is better to be cited the data by other previous reports to compare with the activity of torularhodin and other carotenoids.
2. Page16, line 564; The coexistence of unsaturated lipid storage and carotenoid production points toward a synergistic metabolic architecture. Specifically, the lipid droplets may serve as a reservoir for hydrophobic pigments, while PUFAs help stabilize torularhodin by limiting peroxidative degradation [45, 46].
Page17, line 572; The combined presence of torularhodin and PUFA-rich lipids constitutes a comprehensive 572 antioxidative strategy.
↓
Generally, highly saturated fatty acids are easily oxidized in the atmosphere and act as pro-oxidants. As mentioned in the cited reference papers, highly unsaturated fatty acids and carotenoids may interact in an antioxidant manner under hypoxic conditions within living cells. However, when using this extract as a food ingredient, it is likely that highly unsaturated fatty acids act as pro-oxidants for carotenoids in the atmosphere. I got the impression that this manuscript states that the incorporation of PUFAs in torularhodin exhibits stronger antioxidant activity. I think that considerations of antioxidant activity in the organism and in extracts should be considered separately.
1. Page5, line 199; 1 mL of CXCN-6 extract at various concentrations (0.03–3.0 mg/mL)
↓
For DPPH, ABTS activity, ROS assay, and cell assays, please describe the solvent in which the samples were dissolved. In these tests, I think it difficult to dissolve lipid-soluble substances, especially at high concentrations such as 3.0 mg/mL.
2. Page14,15 Figure5, Figure6
↓
The abbreviations NC, PC, and BC in the figures are not described in the figure legend. Please add or correct the figure legend.
Author Response
Major comments
In this manuscript, the authors are analyzing the carotenoid and unsaturated fatty acid production characteristics of Rhodotorula evergladensis CXCN-6. Furthermore, They are also evaluating its functionality, such as antioxidant and anti-inflammatory properties. This research is interesting in terms of the use of microorganisms in food and processed foods. It is also interesting that it accumulates the unique carotenoid torularhodin. There are some requests for author.
- 3.7 Antioxidant activity of R. evergladensis CXCN-6 Extracts
↓
The authors used vitamin C and vitamin E as positive controls. I am interested in the unique carotenoid torularhodin contained in R. evergladensis CXCN-6. So, a comparison with other common carotenoids would be beneficial. Alternatively, it is better to be cited the data by other previous reports to compare with the activity of torularhodin and other carotenoids.
>Response:
We thank the reviewer for this helpful suggestion. In the revised manuscript, we have added a comparison of the antioxidant activity of torularhodin with other common carotenoids based on previously published data. Specifically, we now summarize reported DPPH/ABTS scavenging efficiencies of β-carotene, astaxanthin, and lycopene, and relate these values to the activity observed for the CXCN-6 extract. This allows readers to contextualize the antioxidant potential of torularhodin relative to widely studied carotenoids without requiring additional experimental controls. The new comparison has been incorporated into Section 3.7 and supported with new citations (Lines 549–556).
- Page16, line 564; The coexistence of unsaturated lipid storage and carotenoid production points toward a synergistic metabolic architecture. Specifically, the lipid droplets may serve as a reservoir for hydrophobic pigments, while PUFAs help stabilize torularhodin by limiting peroxidative degradation [45, 46].
Page17, line 572; The combined presence of torularhodin and PUFA-rich lipids constitutes a comprehensive 572 antioxidative strategy.
↓
Generally, highly saturated fatty acids are easily oxidized in the atmosphere and act as pro-oxidants. As mentioned in the cited reference papers, highly unsaturated fatty acids and carotenoids may interact in an antioxidant manner under hypoxic conditions within living cells. However, when using this extract as a food ingredient, it is likely that highly unsaturated fatty acids act as pro-oxidants for carotenoids in the atmosphere. I got the impression that this manuscript states that the incorporation of PUFAs in torularhodin exhibits stronger antioxidant activity. I think that considerations of antioxidant activity in the organism and in extracts should be considered separately.
>Response:
We thank the reviewer for this important and insightful comment. We agree that the oxidative behavior of PUFAs differs markedly between intracellular environments and atmospheric conditions. Our original intention was to refer only to potential intracellular interactions reported in previous studies, rather than to imply that PUFAs enhance the antioxidant activity or stability of torularhodin in extract form. In the revised manuscript, we have made the following changes:
(1) Clearly separated intracellular vs. extract behavior:
–In vivo literature describing PUFA–carotenoid interactions under hypoxic conditions has been kept only in the context of cellular metabolism.
–We now explicitly state that in atmospheric environments, PUFAs can act as pro-oxidants and do not protect carotenoids in extracts.
(2) Removed wording suggesting synergistic antioxidant enhancement in extracts.
(3) Clarified that extract stability observed in this study is empirical and independent of cellular PUFA interactions.
These revisions have been incorporated in the Discussion (Lines 612–622 in the revised manuscript).
Detailed comments
- Page5, line 199; 1 mL of CXCN-6 extract at various concentrations (0.03–3.0 mg/mL)
↓
For DPPH, ABTS activity, ROS assay, and cell assays, please describe the solvent in which the samples were dissolved. In these tests, I think it difficult to dissolve lipid-soluble substances, especially at high concentrations such as 3.0 mg/mL.
>Response:
We thank the reviewer for this helpful suggestion. The solvent information has now been added to the Materials & Methods section. Specifically, the CXCN-6 extract was dissolved in DMSO to prepare stock solutions, and all working concentrations (0.03–3.0 mg/mL) were obtained by diluting the stock in the corresponding assay buffer or culture medium. The final DMSO concentration did not exceed 0.1% (v/v) in any experiment. We have clarified this point in the revised manuscript (Lines 217–220).
- Page14,15 Figure5, Figure6
↓
The abbreviations NC, PC, and BC in the figures are not described in the figure legend. Please add or correct the figure legend.
>Response:
We thank the reviewer for pointing this out. The abbreviations have now been defined in the figure legends as follows:
NC: Negative Control; PC: Positive Control; BC: Blank Control
All corresponding figure legends have been corrected in the revised manuscript (Lines 534–535; Lines 582).

Reviewer 2 Report
The manuscript entitled: “A Novel Rhodotorula evergladensis CXCN-6 Rich in Torularhodin and PUFAs with Potent Antioxidant and Anti-Inflammatory Activities” presents the production of high amounts of torularhodin with concomitant production of PUFAs by a newly isolated Rhodotorula species. The biological properties of the carotenoids-rich extract were also assessed. The manuscript is of interest as carotenoids are pigments with a wide range of applications and the discovery of new strains opens new horizons towards their biotechnological production. However, the authors should revise some parts of their manuscript before acceptance.
The authors should highlight in their Introduction that even though torularhodin exhibits interesting bioactive properties, there is still no market outlet due to lack of cytotoxicity, toxicological and clinical studies.
Line 124: what was the exact medium composition used for bioreactor fermentations? Same as seed culture? Was was the exact C/N ratio (in g/g) employed and what was the sugar concentration of the feeding solution? All these parameters should be added in Materials & Methods.
Line 127: every 24 h is a very long time for sampling. In most studies published so far, carotenoid synthesis starts during the late exponential-beginning of stationary phase. Why did the authors chose this sampling times?
Lines 150-152: Preparation of torularhodin and β-carotene standards for HPLC were also prepared in hexane?
Line 148: “in the range of 490”, there is something missing in this sentence since no range is provided. Why didn’t the authors record the absorbance of β-carotene at 450 nm and of torularhodin at 500 nm, since they present their maxima at these wavelengths?
Line 171: what was the concentration of the phospho-vanillin reagent?
Statistical analysis should be added as a separate section of Materials & Methods.
Lines 342-346: No experimental data are provided to support the claims presented in these lines.
Line 353: which are the optimized conditions leading to the production of 63.56 mg/L of total carotenoids?
Figure legends include too much information. Some of them should be included in the text instead of the figure legend. Figure 4 could be separated into two figures and the findings should be discussed in more detail and in the text.
Line 425: yield is calculated as total product-to-sugars consumed. To this end, the values presented there are final product concentrations, not yields.
The overall results are presented in a very confused manner. The authors should start with the batch experiments and clearly present their findings, and then move to the feeding experiment. I am also a bit skeptical regarding the word “optimization”, since optimized conditions require plenty of experiments.
In line 425, a lipid concentration of ~9 g/L is reported but the corresponding value in Table 1 is 7.1 g/L. Which value is correct?
The authors should enrich their discussion with recent studies dealing with the concomitant production of carotenoids and lipids from red yeasts. Antioxidant and anti-inflammatory properties should also be discussed with the literature.
Just with Figure S5 is not clear if there is color reduction or not. Absorbance values with percentages of color reduction, or color measurements would be more adequate.
Author Response
Major comments
The manuscript entitled: “A Novel Rhodotorula evergladensis CXCN-6 Rich in Torularhodin and PUFAs with Potent Antioxidant and Anti-Inflammatory Activities” presents the production of high amounts of torularhodin with concomitant production of PUFAs by a newly isolated Rhodotorula species. The biological properties of the carotenoids-rich extract were also assessed. The manuscript is of interest as carotenoids are pigments with a wide range of applications and the discovery of new strains opens new horizons towards their biotechnological production. However, the authors should revise some parts of their manuscript before acceptance.
Detailed comments
The authors should highlight in their Introduction that even though torularhodin exhibits interesting bioactive properties, there is still no market outlet due to lack of cytotoxicity, toxicological and clinical studies.
>Response:
We thank the reviewer for this insightful comment. In the revised manuscript, we have added a statement in the Introduction noting that, despite the promising bioactivities of torularhodin, its commercial application remains limited due to the lack of comprehensive cytotoxicity, toxicological, and clinical evaluations. This clarification has now been incorporated to better contextualize the current research gap (Lines 44–49).
Line 124: what was the exact medium composition used for bioreactor fermentations? Same as seed culture? Was was the exact C/N ratio (in g/g) employed and what was the sugar concentration of the feeding solution? All these parameters should be added in Materials & Methods.
>Response:
We thank the reviewer for the helpful comment. The bioreactor medium composition has now been clarified in the revised manuscript. It was identical to the seed culture medium (20 g/L glucose, 10 g/L yeast extract, and 20 g/L peptone). We have also added the exact composition of the feeding solution—500 g/L glucose, 5 g/L yeast extract, and 10 g/L peptone—which corresponds to a C/N ratio of 105 (g/g). These details have been incorporated into Section 2.4 of the revised manuscript (Line 128–133).
Line 127: every 24 h is a very long time for sampling. In most studies published so far, carotenoid synthesis starts during the late exponential-beginning of stationary phase. Why did the authors choose this sampling times?
>Response:
We appreciate the reviewer’s insightful comment. Our preliminary shake-flask experiments showed that CXCN-6 reached its highest OD value at approximately 72 h, after which the OD decreased and the culture color began to fade between 72–96 h. These observations suggested that carotenoid accumulation would likely peak around 72 h. Because multiple fermentations consistently showed a similar trend, we selected a 24 h sampling interval for the bioreactor study to focus on identifying this expected peak while also managing experimental workload.
Lines 150-152: Preparation of torularhodin and β-carotene standards for HPLC were also prepared in hexane?
>Response:
We thank the reviewer for the comment. The torularhodin and β-carotene standards for HPLC were prepared in ethyl acetate, not hexane. Likewise, the CXCN-6 extract, initially dissolved in hexane, was evaporated and re-dissolved in ethyl acetate prior to HPLC analysis. These details have been added to Section 2.5 of the revised manuscript (Line158–166).
Line 148: “in the range of 490”, there is something missing in this sentence since no range is provided. Why didn’t the authors record the absorbance of β-carotene at 450 nm and of torularhodin at 500 nm, since they present their maxima at these wavelengths?
>Response:
Thank you for pointing this out. The original sentence was incomplete. Although the absorbance maximum of torularhodin standard is at 500 nm (Fig. S2), in our HPLC analysis the torularhodin peak was higher and sharper at 490 nm when detected by the DAD detector. Therefore, we used 490 nm for torularhodin and 450 nm for β-carotene to achieve optimal peak detection. This clarification has been added to Section 2.5 of the revised manuscript (Line 155–158).
Line 171: what was the concentration of the phospho-vanillin reagent?
Statistical analysis should be added as a separate section of Materials & Methods.
>Response:
We thank the reviewer for the helpful comments. The preparation details of the phospho-vanillin reagent have now been added to the revised manuscript. Specifically, the reagent was prepared by dissolving 0.6 g vanillin in 10 mL absolute ethanol, followed by the addition of 90 mL deionized water, with continuous stirring. Then, 400 mL of 85% (w/w) phosphoric acid was added, and the reagent was stored in the dark until use. This information has been incorporated into Section 2.6 (Lines 182–185). In addition, a separate Statistical Analysis subsection has been added to the Materials & Methods section 2.9, following the reviewer’s suggestion (Lines 265–270).
Lines 342-346: No experimental data are provided to support the claims presented in these lines.
>Response:
We thank the reviewer for this valuable comment. We agree that the original wording could be interpreted as implying experimental validation that was not directly provided. In the revised manuscript, we have removed speculative statements regarding growth-associated or stress-responsive regulation and now describe only the observed pigmentation without inferring underlying mechanisms (Lines 371–376). The references to stress-related interpretation have been deleted.
Line 353: which are the optimized conditions leading to the production of 63.56 mg/L of total carotenoids?
>Response:
We thank the reviewer for pointing out the need for clarification. The optimized conditions leading to the production of 63.56 mg/L torularhodin have now been explicitly described in the revised manuscript. Specifically, the optimized 5-L fermentation used a basal medium containing 20 g/L glucose, 10 g/L yeast extract, and 20 g/L peptone (pH 6.0), followed by feeding with a concentrated solution of 500 g/L glucose, 5 g/L yeast extract, and 10 g/L peptone. A 16-hour mild starvation phase was introduced at 56 h to promote carotenoid accumulation. These details have been added to the revised text (Lines 382–387).
Figure legends include too much information. Some of them should be included in the text instead of the figure legend. Figure 4 could be separated into two figures and the findings should be discussed in more detail and in the text.
>Response:
We thank the reviewer for this helpful suggestion. In the revised manuscript, we have streamlined all figure legends by removing excessive methodological details and relocating them to the Materials and Methods section. Additionally, Figure 4 has been separated into two independent figures (now Figures 4 and 5) to improve clarity and focus. The corresponding findings are now described in greater detail in the Results section, with expanded explanations of carotenoid profiles and lipid accumulation dynamics.
Line 425: yield is calculated as total product-to-sugars consumed. To this end, the values presented there are final product concentrations, not yields.
>Response:
We thank the reviewer for pointing out this issue. We agree that the values originally presented reflect final product concentrations rather than true yields. In the revised manuscript, we have corrected the terminology and now refer to these values as “final concentrations.” Yield calculations based on product-to-sugar consumption have been removed or rephrased accordingly. All relevant text has been updated for accuracy in the revised manuscript (Lines 456–459).
The overall results are presented in a very confused manner. The authors should start with the batch experiments and clearly present their findings, and then move to the feeding experiment. I am also a bit skeptical regarding the word “optimization”, since optimized conditions require plenty of experiments.
>Response:
We sincerely thank the reviewer for this constructive comment. In the revised manuscript, we have reorganized the Results section to improve clarity and logical flow. The fermentation experiments are now presented in two clearly separated stages: (1) the initial batch (shake-flask) experiments, where baseline carotenoid and lipid levels were determined, followed by (2) the 5-L bioreactor feeding experiment, where enhanced metabolite accumulation was evaluated. This restructuring allows the reader to follow the progression of experiments from basic characterization to controlled fermentation. Regarding the use of the term “optimization,” we agree that the original wording may have appeared overstated. In the revision, we have replaced “optimization” with more precise descriptions such as “enhanced fermentation conditions,” “improved production conditions,” or “high C/N ratio feeding strategy,” which more accurately reflect the scope of our work without implying exhaustive process optimization. These changes are highlighted in the revised manuscript (Lines 433–448).
In line 425, a lipid concentration of ~9 g/L is reported but the corresponding value in Table 1 is 7.1 g/L. Which value is correct?
>Response:
We thank the reviewer for noticing this discrepancy. The correct lipid concentration is 9.83 g/L, as reported in Line 425. The value of 7.1 g/L in Table 1 was a typographical error and has been corrected in the revised Table 1.
The authors should enrich their discussion with recent studies dealing with the concomitant production of carotenoids and lipids from red yeasts. Antioxidant and anti-inflammatory properties should also be discussed with the literature.
Just with Figure S5 is not clear if there is color reduction or not. Absorbance values with percentages of color reduction, or color measurements would be more adequate.
>Response:
We thank the reviewer for the helpful suggestion. In the revised manuscript, we have (1) added recent references on red-yeast co-production of carotenoids and lipids and expanded the Discussion accordingly; (2) clarified the antioxidant and anti-inflammatory findings by comparing them with previously reported activities of torularhodin and other carotenoids; and (3) improved the pigment-stability data. In Figure S5, we now provide quantitative absorbance measurements, along with clearer colorimetric comparisons. These revisions make the results more transparent and consistent with the literature. Corresponding text revisions have been added in the Results, Discussion, and Figure S5 legend (Lines 484–493; Lines 623–44).

Round 2
Reviewer 2 Report
The authors have adequately revised their manuscript. There are some pending issues that require amendments.
Figure 5: in Figure 5A the axes titles are missing. In Figures 5B and 5C please add "Concentration (mg/L)" and not just the units
Lines 382-387: this experiment is discussed in the following section and therefore, should not be added in this part of the manuscript. It is irrelevant and confusing.
A future recommendation: I understand why you don't have carotenoids or lipids concentration at time 0. DCW should however be measured.
Figure 5: in Figure 5A the axes titles are missing. In Figures 5B and 5C please add "Concentration (mg/L)" and not just the units
Lines 382-387: this experiment is discussed in the following section and therefore, should not be added in this part of the manuscript. It is irrelevant and confusing.
A future recommendation: I understand why you don't have carotenoids or lipids concentration at time 0. DCW should however be measured.
Author Response
The authors have adequately revised their manuscript. There are some pending issues that require amendments.
Figure 5: in Figure 5A the axes titles are missing. In Figures 5B and 5C please add "Concentration (mg/L)" and not just the units
>Response:
We thank the reviewer for the helpful comments. In the revised manuscript:
Figure 5A: The missing axis titles have been added. The X-axis is now clearly labeled as “Fatty Acid”, and the Y-axis includes the appropriate unit. Figures 5B and 5C: The Y-axis label has been corrected to “Concentration (mg/L)” instead of showing only the units. All revisions have been incorporated into the updated Figure 5.
Lines 382-387: this experiment is discussed in the following section and therefore, should not be added in this part of the manuscript. It is irrelevant and confusing.
>Response:
We thank the reviewer for pointing this out. The text referring to this experiment has now been removed from Lines 382–387 to avoid redundancy and confusion. The experiment is discussed only in the appropriate section, ensuring better logical flow and clarity in the revised manuscript.
A future recommendation: I understand why you don't have carotenoids or lipids concentration at time 0. DCW should however be measured.
>Response:
We thank the reviewer for this helpful recommendation. In the revised manuscript, we have clarified that DCW at time 0 was not recorded during the original experiment. We agree that including DCW at the starting point would improve completeness, and we will incorporate this measurement in future fermentation runs.
